# Molecular Rescue of Dyrk1A Overexpression Alterations in Mice with Fontup^®^ Dietary Supplement: Role of Green Tea Catechins

**DOI:** 10.3390/ijms21041404

**Published:** 2020-02-19

**Authors:** Yuchen Gu, Gautier Moroy, Jean-Louis Paul, Anne-Sophie Rebillat, Mara Dierssen, Rafael de la Torre, Cécile Cieuta-Walti, Julien Dairou, Nathalie Janel

**Affiliations:** 1Université de Paris, BFA, UMR 8251, CNRS, F-75013 Paris, France; guyuchen0509@hotmail.com; 2Université de Paris, BFA, UMR 8251, CNRS, ERL U1133, Inserm, F-75013 Paris, France; gautier.moroy@univ-paris-diderot.fr; 3Department of Biochemistry, Georges Pompidou European Hospital, Assistance Publique-Hôpitaux de Paris (AP-HP), F-75013 Paris, France; jean-louis.paul@egp.aphp.fr; 4Institut Jérôme Lejeune, F-75013 Paris, France; annesophie.rebillat@institutlejeune.org (A.-S.R.); cecile.cieuta-walti@institutlejeune.org (C.C.-W.); 5Centre for Genomic Regulation (CRG), The Barcelona Institute of Science and Technology, 08003 Barcelona, Spain; mara.dierssen@crg.eu; 6Universitat Pompeu Fabra (UPF), 08003 Barcelona, Spain; rtorre@imim.es; 7Integrative Pharmacology and Systems Neuroscience Research Group, Neurosciences Research Program, IMIM (Hospital del Mar Medical Research Institute), 08003 Barcelona, Spain; 8Université de Paris, Laboratoire de Chimie et Biochimie Pharmacologiques et Toxicologique, UMR 8601, CNRS, F-75013 Paris, France; julien.dairou@univ-paris-diderot.fr

**Keywords:** DYRK1A, docking, EGCG, ECG, RTK pathway, BDNF

## Abstract

Epigallocatechin gallate (EGCG) is an inhibitor of DYRK1A, a serine/threonine kinase considered to be a major contributor of cognitive dysfunctions in Down syndrome (DS). Two clinical trials in adult patients with DS have shown the safety and efficacy to improve cognitive phenotypes using commercial green tea extract containing EGCG (45% content). In the present study, we performed a preclinical study using FontUp^®^, a new nutritional supplement with a chocolate taste specifically formulated for the nutritional needs of patients with DS and enriched with a standardized amount of EGCG in young mice overexpressing *Dyrk1A* (TgBACDyrk1A). This preparation is differential with previous one used, because its green tea extract has been purified to up 94% EGCG of total catechins. We analyzed the in vitro effect of green tea catechins not only for EGCG, but for others residually contained in FontUp^®^, on DYRK1A kinase activity. Like EGCG, epicatechin gallate was a noncompetitive inhibitor against ATP, molecular docking computations confirming these results. Oral FontUp^®^ normalized brain and plasma biomarkers deregulated in TgBACDyrk1A, without negative effect on liver and cardiac functions. We compared the bioavailability of EGCG in plasma and brain of mice and have demonstrated that EGCG had well crossed the blood-brain barrier.

## 1. Introduction

DYRK1A (dual-specificity tyrosine phosphorylation-related kinase 1A) is a dual-specificity protein kinase which gene is located on 21q22.2 of human chromosome 21 [1] overexpressed in Down syndrome (DS). DS or trisomy 21, affects 1 in 800 live births, being the most common genetic developmental disorder. Phenotypic characteristics are complex and variable, but reduced learning and memory capacities are common to all individuals with DS [2]. They exhibit impaired development of the nervous system and a delay in cognitive development leading to intellectual disability [2]. Among the candidate genes for cognitive impairment in DS, *DYRK1A* has been mainly studied due to its role in neurodevelopment [3] and on the DS brain defects. A large number of studies have evaluated the effect of normalizing the DYRK1A expression and/or kinase activity on the learning and memory capacities of DS mouse models [4]. There are several compounds with inhibitory activity for DYRK1A, but only a few have been tested in preclinical studies. Some have been isolated from natural sources. Among them, the green tea catechin, epigallocatechin-3-gallate (EGCG) is the most used in preclinical studies, considering the safety of the molecule and the interesting potency to inhibit DYRK1A activity with IC50 of 330 nM [5]. Most of the DYRK1A inhibition studies with EGCG treatment showed some beneficial effects [6,7,8,9,10,11,12,13,14,15].

In light of these encouraging preclinical data, DYRK1A inhibition has indeed become a viable option in the clinical setting. Two clinical trials in adult patients with DS have evaluated the safety and efficacy of EGCG, the only DYRK1A inhibitor tested until now in humans. In a randomized, double blind study, 31 young adults with DS (aged 14 to 29 years) received a green tea extract (oral dose of 9 mg/kg/day, Mega Green Tea Extract, Lightly Caffeinated, enriched up to 45% with EGCG, Life Extension^®^, Fort Lauderdale, FL, USA) or placebo treatments over a period of three months. The study established a favorable safety profile of EGCG treatment in young individuals with DS, with no alteration of the hepatic function together with an improvement of lipid profile (including total and LDL cholesterol) [8]. EGCG improved visual recognition memory, working memory performance, psychomotor speed and social functioning. In a double-blind, randomized, placebo-controlled, phase 2 clinical trial (TESDAD study), the safety and efficacy of the same green tea extract supplementation paired with cognitive training was compared to cognitive training alone. 84 young adults with DS aged from 16 to 34 years old were enrolled and assigned to cognitive training alone or cognitive training with EGCG supplementation for 12 months with a follow-up at 6-month post-treatment period. As in the first clinical study, no changes were observed in hepatic functionality. The combined EGCG treatment with cognitive training improved certain areas of the memory, executive functions and competences in everyday life. Neuroimaging analysis revealed improvements in functional connectivity and normalization of cortical excitability when combined EGCG treatment and cognitive training [16]. The results also suggest that some of the changes observed persist over at least six months after discontinuing the treatment. One main limitation with these studies is that the concentration of EGCG in green tea extract was 45%, and other various polyphenols, in addition to EGCG, may have a role on DYRK1A inhibition.

FontUp^®^ is a new nutritional supplement with a chocolate taste, enriched with a standardized amount of EGCG (510 mg/100 g), which is now being used in ongoing clinical trials, and also by individuals having DS. In this study, we analyzed the in vivo effect of FontUp^®^ administration on different biological markers deregulated in mice overexpressing *Dyrk1A.* We have also compared the in vitro effect of green tea phenolic compounds (EGCG, epigallocatechin (EGC), epicatechin gallate (ECG) and epicatechin (EC) on DYRK1A kinase activity. 

## 2. Results

### 2.1. Quantification of FontUp^®^ Polyphenols

We first monitored the amount of polyphenols in the FontUp^®^ formulation by HPLC coupled with DAD. The retention times for the elution of standards EGCG, ECG, EGC and EC were 15.5, 19.7, 10.6 and 16.7 min, respectively. Overall, a good resolution was achieved for these four catechins. Figure 1 shows superposition of two chromatograms at 280 nm, the blue one is a mix of eleven standards of catechins and the red is the chromatogram of the methanolic extract of FontUp^®^ (Figure 1). The most abundant peak in Fontup^®^ sample has a retention time of 15.5 min which overlaps with the peak corresponding to the EGCG. A quantitative analysis reveals that the concentration of EGCG in FontUp^®^ is 1.61 mg/mL, therefore comparable to that given by the manufacturer. We also observe, to a lesser amount, the presence of EC and ECG at 0.1 mg/mL. Therefore, we demonstrate that there is essentially EGCG (94% of total catechins) in FontUp^®^. 

### 2.2. In Vitro Inhibition and Molecular Docking of DYRK1A

EGCG has been identified as a non-competitive inhibitor against both substrates of DYRK1A [17]. To test whether the inhibitory properties of DYRK1A are also found with other polyphenols contained in FontUp^®^, we tested DYRK1A kinase inhibition with different concentrations of each compound. Table 1 and Table 2 show DYRK1A -ΔC activity and DYRK1A (DYRK1A native human protein, PV3785, Thermo Fisher Scientific, Waltham, MA, USA) activity, respectively, as a function of the concentration of each compound for 1 mM of ATP. As expected, the DYRK1A-ΔC and DYRK1A activities decreased with increasing concentrations of EGCG (Table 1 and Table 2). 

The DYRK1A-ΔC activity was also decreased in presence of increased concentrations of ECG (Table 1), this result being confirmed by the analysis of DYRK1A activity (Table 2). No effect was found for EGC and EC (Table 1). EGCG is the ester of epigallocatechin and gallic acid. We can conclude that there is no effect of the hydroxyl group, but an important effect of gallic acid. In parallel, we measured the DYRK1A-ΔC and DYRK1A activities with ECG or EGCG in presence of different concentrations of ATP (200–800 µM) and found no difference (Table 3 and Table 4). Therefore, we determined that, like EGCG, ECG is a noncompetitive inhibitor of DYRK1A. 

Molecular docking is a bioinformatics tool which is able to predict the position and orientation of a molecule at a protein surface. We used it in order to understand how the tested compounds interact with DYRK1A, namely, where their binding site on DYRK1A is and which residues are involved in the interaction. The EGCG docked conformations with the lowest binding energy are mainly localized in the catalytic site and in a flat pocket centered on residue L457 (L457 allows to locate the pocket, but has no interaction) (Figure 2A). The predicted binding energy for the two poses is equal to −9.6 kcal·mol^−1^, suggesting that EGCG has the same affinity for these two sites. However, the docked conformations in the catalytic site are not in agreement with the experimental measurements showing that the EGCG inhibitory activity is non-competitive. In other words, EGCG does not compete with ATP to bind in the catalytic site. This pose is therefore not correct. The EGCG binding in the catalytic site is likely energetically favorable due to shape of the catalytic site: it is deep and large, which may induce overestimation of the hydrophobic and/or steric interaction terms by the scoring function.

On the other hand, the pose in the flat pocket seems to be more suitable. It is in accordance with the experimental data, explaining the non-competitive inhibitory activity of EGCG. Several non-bonded interactions stabilized the interaction (Figure 2B). One cation-π interaction with K222 and three hydrogen bonds involving H424, R458 and Y462 are formed between DYRK1A and EGCG. The ECG interacts in a similar binding mode to the one of EGCG (Figure 2C), with a slightly lower predicted energy, −9.8 kcal·mol^−1^. The same three hydrogen bonds and the same cation-π interaction observed for EGCG are detected for EGC interaction with DYRK1A. The best EC and EGC binding mode, is associated to a predicted energy of −7.9 Kcal·mol^−1^, suggesting that EC and EGC have a low affinity for DYRK1A. The molecular docking computations are therefore consistent with the experimental results. 

### 2.3. Effects of Low, Intermediate and High Doses of FontUp^®^ Administration on Plasma Biomarkers Linked with DYRK1A 

In order to determine the dose which will show the less toxicity in mice, we first administered in wild type (WT) mice three different doses of FontUp^®^ based on previous results showing beneficial effects on biomarkers linked with DYRK1A after only two days of harmine treatment, a DYRK1A inhibitor [18]. The dosing was 25 mg/Kg of FontUp^®^ for low dose (D1, Table 5), 50 mg/Kg for intermediate dose (D2, Table 5), and 75 mg/Kg for high dose (D3, Table 5). To avoid factors such as circadian regulation of drinking and pharmacokinetic differences in absorption, distribution, and metabolism due to the daily pattern of fluid consumption of the mouse, we administered FontUp^®^ by oral gavage. 

EGCG plasma levels were correlated with the administered doses (Table 5). Plasma homocysteine (hcy) levels were previously found to be decreased in mice overexpressing *Dyrk1A* [8,19]. We detected an increase in plasma hcy concentrations only for the intermediate (D2) EGCG dose, with an increase in glutathione (GSH) level, an antioxidant molecule (Table 5). 

In plasma of TgBACDyrk1A mice we first confirmed the decreased concentrations of hcy (Figure 3A) and Alanine aminotransferase (ALT) (Figure 3B) [20]. We found increased plasma hcy concentrations in TgBACDyrk1A mice with the D2 dose (Figure 3A), without statistical effect for plasma ALT levels (Figure 3B). We also analyzed plasma fibronectin, a glycoprotein identified as DYRK1A-interacting protein in a previous yeast two-hybrid random-primed human adult brain cDNA library screening (personal communication). Fibronectin levels were increased in plasma of TgBACDyrk1A mice (Figure 3C), this increase being counteracted by the D2 EGCG dose (Figure 3C).

### 2.4. Effects of the Higher Dose of FontUp^®^ Administration on Liver and Cardiac Biomarkers

Previous studies demonstrated that high doses of EGCG induce cardiac fibrosis and acute liver failure [21,22]. Recently we showed that Dyrk1A overexpression protects liver function [20]. Therefore, we analyzed plasma ALT levels to verify whether its inhibition had no deleterious effect on liver function. We found a decrease in plasma ALT levels upon FontUp^®^ treatment, D2 being the most effective dose (Table 5). We also previously found that inhibition of DYRK1A induces cardiac dysfunction [23]. Therefore, we analyzed plasma galectin-3 levels, as a surrogate biomarker and found no effect of any of the three doses suggesting no cardiac toxicity of FontUp^®^ treatment (Table 5). 

Even though these initial experiments were not suggestive of toxicity, in order to ensure whether FontUp^®^ treatment had no deleterious effects on liver and cardiac function, WT and TgBACDyrk1A mice were force-fed with the higher dose (D3) during eight days [22]. FontUp^®^ D3 administration did not modify plasma ALT levels in WT or TgBACDyrk1A (Figure 4A), but significantly reduced galectin-3 levels in WT mice (Figure 4B). In the heart tissue, Collagen I and III, markers of fibrosis, were decreased (Figure 5A,B), and αSMA, a marker of cardiac hypertrophy, was increased in TgBACDyrk1A mice, compared to WT (Figure 5C). After FontUp^®^ D3 treatment, levels of collagen I and III were decreased in heart of WT mice but were not modified in TgBACDyrk1A (Figure 5A,B). FontUp^®^ treatment significantly reduced heart αSMA levels in TgBACDyrk1A but no treatment effect was found for WT (Figure 5C). 

### 2.5. Effects of the Intermediate Dose of FontUp^®^ Administration on Brain Biomarkers Linked with DYRK1A

To determine whether FontUp^®^ has positive effect on brain biomarkers linked with DYRK1A inhibition we used the intermediate dose D2 (50 mg/Kg of FontUp^®^) based on the results obtained in plasma of mice. We first demonstrated that EGCG can be detected in plasma and brain tissue using mass spectral analysis. The bioavailability was analyzed by measuring EGCG levels in plasma and brain of WT and TgBACDyrk1A mice in order to assess whether it had likely crossed the blood-brain barrier (BBB). Compared to WT mice, EGCG concentrations were decreased in plasma and increased in brain of TgBACDyrk1A mice, and the partition coefficient (Kp = brain concentration/plasma concentration) demonstrates that EGCG had crossed the BBB (Table 6).

No effect of FontUp^®^ treatment was found on *Dyrk1A* mRNA and DYRK1A protein levels in brain of WT and TgBACDyrk1A mice (data not shown). We confirmed previous observations that BDNF concentrations were decreased [24] while phospho-Erk (PErk) and phospho-Akt (PAkt) were increased [18] in brain of TgBACDyrk1A (Figure 6A–C). 

A beneficial effect was found not only on BDNF level (Figure 6A), which was significantly increased, but also on the activation of receptor tyrosine kinase (RTK) in brain of TgBACDyrk1A mice after FontUp^®^ D2 treatment, as shown by the normalization of erk and Akt phosphorylation (Figure 6B,C) [25]. Ring Finger Protein 216 (RNF216) (Figure 7A) and IkBα levels (Figure 7B) increase in brain of TgBACDyrk1A, as previously described [26]. FontUp^®^ induced an increase of RNF216 concentration in brain of WT (Figure 7A) but no significant changes in the case of TgBACDyrk1A mice. Moreover, no effect was found for IkBα concentrations (Figure 7B). However, a positive correlation was found between RNF216 and IkBα concentrations (*r* = 0.56, *p* < 0.01), suggesting a beneficial effect of the EGCG D2 dose on the NFkB pathway.

## 3. Discussion 

In this report, we have evaluated the safety profile of a currently widely used dietetic preparation used by DS individuals containing a purified green tea extract with a concentration of 94% EGCG of total catechins (FontUp^®^) as compared with previous preclinical and clinical studies performed with green tea extracts with a 45% EGCG content). Results obtained will essentially allow to determine effects of EGCG over the background of other catechins present in former green tea extracts. The preparation tested has been shown to be safe as biomarkers of hepatotoxicity and cardiotoxicity remain unaltered within the tested dose range of EGCG (25–75 mg/kg). We have also demonstrated that EGCG is distributed in brain. Plasma biomarkers suggest that EGCG-elicited effects are related with DYRK1A interaction (e.g., hcy, fibronectin). In the same way several brain biomarkers, support that beneficial effects derived from EGCG are the result of its interaction with DYRK1A (e.g., BDNF, NFkB). Studies with other green tea catechins like ECG suggest that they may contribute (to a lesser extent than EGCG) to biological effects of non-purified green tea extracts used previously in preclinical and clinical studies. 

Cognitive impairment in individuals with DS represents the major concern for families and society. Several preclinical studies have shown that in mouse models of DS it is possible to improve or even rescue the major neurodevelopmental alterations [27]. This is new hope for therapeutic interventions in individuals with DS. Many studies aimed at deciphering whether it is possible to correct phenotype through modulation of DYRK1A activity. Preclinical studies performed in mouse models of DS have reported improved behavioral outcomes with EGCG, an inhibitor of DYRK1A [6,7,8,9,10,11,12,13,14,15]. Two clinical trials in adult patients with DS have evaluated the safety and efficacy to improve cognitive phenotypes of green tea extracts containing 45% EGCG [8,16]. FontUp^®^ is a new nutritional supplement with a chocolate taste, enriched with EGCG (94% of total catechins, Figure 1). It has been specifically formulated for the nutritional needs of patients with DS.

It is interesting to note that Ts65Dn mice treated with pure EGCG treatment in drinking water with two different doses (20 mg/kg/day or 50 mg/kg/day) did not show behavioral improvements [12,14]. Pure EGCG treatment vs. EGCG in combination with other green tea extracts can account for these discrepancies. In this sense, it has been demonstrated that the EGCG dose and composition of EGCG-containing supplements is important in correcting skeletal deficits in Ts65Dn mice, the presence of other catechins found in green tea extract accounting for the different effects on bones [28]. We also observe, a lower presence of EC and ECG in Fontup^®^. Therefore, we have analyzed the in vitro effect of these compounds on DYRK1A activity and found DYRK1A inhibition with ECG, like EGCG. In vitro analysis emphasizes the role of gallic acid on DYRK1A inhibition. Moreover, ECG, like EGCG, was noncompetitive against ATP. In silico results demonstrate that ECG interacts in a similar binding mode than EGCG and therefore are also consistent with in vitro results. Taken together, our results demonstrate the same effect of EGCG and ECG on DYRK1A inhibition, without antagonistic effect of EC.

In terms of toxicity, no deleterious effect of the higher dose of FontUp^®^ was found on plasma ALT and Galectin-3 levels in WT nor in TgBACDyrk1A, demonstrating the safety on liver and heart function. Galectin-3 is a biomarker of cardiovascular fibrosis and disease [29], and ALT is an important biomarker for diagnosing liver disease and reflecting liver damages. Previous results have elucidated the potential antifibrogenic role of green tea catechins, like EGCG, in a CCl_4_-induced fibrotic rat showing an increased serum ALT [25]. They postulated that EGCG are potential therapeutic candidates in antifibrotic therapy [30]. Excessive fibrosis in the extracellular matrix with overactivated myofibroblasts increases myocardial stiffness resulting in myocardial dysfunction. In our study, no deleterious effect was found with the higher dose of FontUp^®^ on heart collagen I and III, markers of matrix protein production, in the heart of WT and TgBACDyrk1A. Moreover, a beneficial effect on αSMA, marker of myofibroblast activation, was found in heart of TgBACDyrk1A. A number of studies suggest that consumption of green tea decreased the risk of cardiovascular diseases. Previous studies demonstrated that green tea catechins might be useful compounds for the development of therapeutic agents to treat hypertrophic cardiomyopathy [31,32], due to antioxidative and anti-inflammatory effects [32,33]. 

We also demonstrated beneficial effects of intermediate dose of FontUp^®^ on plasma and brain biomarkers deregulated in mice overexpressing *Dyrk1A*. Previous studies have shown that EGCG can cross the BBB, even at a very low concentration, by the use of a BBB model kit and in conscious and freely moving rat [34,35,36]. In WT mice the presence of catechins from Mega Green Tea Extract was first assessed in plasma [15]. In our study, EGCG is well detected not only in plasma but also in brain of WT and TgBACDyrk1A mice. Moreover, we found that in mice overexpressing *Dyrk1A*, brain EGCG concentration is higher than plasma EGCG concentration. These results indicate that, unlike WT mice, brain delivery is less limited in mice overexpressing *Dyrk1A*.

In plasma, increased hcy level was found after FontUp^®^ treatment, as previously described [8], but below the threshold value defining moderate hyperhomocysteinemia and associated with a positive effect on ALT levels. Fibronectin has been identified as a potential DS biomarker in the maternal serum of pregnant women carrying DS fetus [37]. We found that plasma fibronectin levels are increased in TgBACDyrk1A mice compared to WT mice, and FontUp^®^ administration decreases it, even if the difference is not significant in case of WT mice. BDNF is a neurotrophin that plays an important role in neuronal survival and in processes of functional and structural synaptic plasticity [38]. Previous results have already demonstrated the beneficial effect of polyphenol-based diets on BDNF mRNA level in another model of mice overexpressing *Dyrk1A* [6]. Another study has investigated the use of a pharmacological approach indented to promote BDNF in Ts65Dn mice, a mouse model of DS, and provide evidence that the BDNF-mimetic drug 7,8-dihydroxyflavone rescued learning and memory impairments [39]. We here demonstrate the beneficial effect of FontUp^®^ administration on BDNF protein level in TgBACDyrk1A. The studies on transgenic mice overexpressing *Dyrk1A* established a relationship between brain DYRK1A protein and BDNF-dependent TrkB pathway as well as RTK-mediated signaling pathways such as MAPK/ERK and PI3K/Akt pathways [18]. We also demonstrate the beneficial effect of FontUp^®^ administration on MAPK/ERK and PI3K/Akt pathways normalization in brain of mice overexpressing *Dyrk1A*. Many studies indicate that RTKs are one of the critical targets of green tea catechins, especially EGCG [40]. Therefore, EGCG can inhibit the MAPK/ERK and PI3K/Akt signaling pathways, as well as the activation of NFkB. In this way, EGCG has been shown to inhibit IκB kinase activity, thereby IkBα phosphorylation and degradation and NFkB activation in many different cell types [41]. It is interesting to note that this inhibition of IKK activity by EGCG is related to the presence of the gallate group because the polyphenols lacking gallate group did not inhibit IKK activity [42]. We found again that not only IkBα but also RNF216 levels were always increased in brain of mice overexpressing *Dyrk1A* after FontUp^®^ administration, which emphasizes the beneficial effect of this treatment on NFkB pathway inhibition. Note that EGCG also upregulates RNF216 expression in macrophages [43]. 

## 4. Materials and Methods 

### 4.1. Quantification of FontUp^®^ Polyphenols

FontUp^®^ was prepared according to the manufacturer’s recommendations (40 g of FontUp^®^ powder was dissolved in 100 mL of distilled water and strongly mixed at room temperature). Successive dilutions were done with distilled water up to 0.4 mg/mL. Then, a methanol extraction was done and analyzed by reverse phase high performance liquid chromatography (HPLC). The extract was injected onto a C_18_ column (Kromasil 250 × 4.6 mm, 5 µm) to determine FontUp^®^ phenolic compounds. Briefly, the two mobile phases used for gradient HPLC elution were (A) 0.1% perchloric acid in water and (B) methanol. The flow rate was set at 1 mL/min. The gradient elution profile was 5% B (isocratic) for 11 min, B was gradually increased to 40% at 31 min, to 80% at 40 min and back to 5% until 45 min. The injection volume was 20 µL. Each sample was filtered through a 0.22 µm PTFE membrane filter. The eluent was monitored using a diode array detector (DAD). Quantification was carried out using the external standard method. Chromatographic peaks were identified in the samples by comparing their retention times and UV spectra with those of reference standards and by co-chromatography with added standards. Quantification was performed from the peak area of each component and its interpolation to the corresponding calibration curve.

### 4.2. Experimental Mice Model

Mice carrying the murine BAC containing one copy of *Dyrk1A* (TgBACDyrk1A) were maintained on a C57Bl/6J background and genotyped as described [44]. Briefly, the murine bacterial artificial chromosome 189 N3 (mBACtgDyrk1A) strain was constructed by electroporating HM-1 embryonic stem cells with the retrofitted BAC-189N3 [44]. WT littermates were used as controls. Mice were housed in a controlled environment with *ad libitum* access to food and water on a 12-h light/dark cycle. The number of mice used and animal suffering were minimized as much as possible. Male mice from the same litter, two months of age, were used. All procedures were carried out in accordance with the ethical standards of French and European regulations (European Communities Council Directive, 86/609/EEC). Official authorization from the French Ministry of Agriculture was granted to perform research and experiments on animals (authorization number 75–369) and the experimental protocol was approved by the institutional animal care and use committee of the Paris Diderot University (CEEA40) (17 December 2015).

### 4.3. FontUp^®^ Treatment

FontUp^®^ was prepared daily as described above and strongly mixed at room temperature. Male mice received an oral gavage of three different doses (25, 50 and 75 mg/Kg), in the morning and in the evening for two or eight days. The last dose was administered 4 h before sacrifice.

### 4.4. Preparation of Serum Samples, Tissue Collection, and Plasma Assays

Blood samples were obtained by retro-orbital sinus sampling with heparinized capillaries, collected into tubes containing a 1/10 volume of 3.8% sodium citrate, and immediately placed on ice. Plasma was isolated by centrifugation at 2500× *g* for 15 min at 4 °C. 80 µL of plasma samples were mixed with 20 µL 0.2 M NaH_2_PO_4_ solution containing 20% ascorbic acid and 0.1% EDTA, pH 3.6 for polyphenol assays. Plasma samples were store at −80 °C until use. Brains and hearts were harvested, snap-frozen in liquid nitrogen, and stored at −80 °C until use. Plasma total hcy, defined as the total concentration of hcy after quantitative reductive cleavage of all disulfide bonds, and total GSH were assayed using a fluorometric HPLC method previously described [45]. ALT concentrations were determined using the Alanine Aminotransferase Activity Assay Kit (Sigma-Aldrich, Saint-Quentin Fallavier, France), based on the pyruvate generated. Plasma galectin-3 (Abcam ELISA kit, Paris, France), and fibronectin (Abcam ELISA kit) were assessed using sandwich ELISA. After removal of unbound conjugates, bound enzyme activity was assessed by use of a chromogenic substrate for measurement at 450 nm by a microplate reader (Flex Station 3, Molecular Devices, Ltd., Wokhingham, UK). 

### 4.5. Protein Extraction and Analysis

Total protein samples were prepared by homogenizing brain and heart in 500 µL phosphate-buffered saline (PBS) with a cocktail of protease inhibitors. Protein concentrations were determined with the Bio-Rad Protein Assay reagent (Bio-Rad, Hercules, CA, USA). To assess the relative amount of proteins, we used a slot blot method after testing the specificity of antibodies by western blotting. Protein preparations were blotted on a Hybond-C Extra membrane (GE Healthcare Europe GmbH) using a Bio-Dot SF Microfiltration Apparatus (Bio-Rad). After transfer, membranes were saturated by incubation in 10% *w*/*v* nonfat milk powder or 5% *w*/*v* BSA in Tris-saline buffer (1.5 mM Tris base, pH 8; 5 mM NaCl; 0.1% Tween-20), and incubated overnight at 4 °C with an antibody directed against Akt1/2/3 (1/1000; Santa Cruz Biotechnology, Tebu, France), Phospho-Akt1/2/3 (Ser 473; 1/1000; Santa Cruz Biotechnology), collagen III (1/1000, Abcam), collagen I (1/1000, Abcam), IkBα (1/1000, Cell Signaling Technology, Danvers, MA, USA), p44/42 MAPK (ERK1/2)(137F5) (1/2000; Cell Signaling Technology, Ozyme, France), Phospho-p44/42 MAPK (ERK1/2) (Thr202/Tyr204) (1/2000; Cell Signaling Technology), RNF216 (1/2000, Abcam), α smooth muscle actin (SMA) (1/400, Abcam). Binding of the primary antibody was detected by incubation with horseradish peroxidase (HRP)-conjugated secondary antibody using the Western Blotting Luminol Reagent (Santa Cruz Biotechnology). Ponceau-S coloration (Sigma-Aldrich, Saint-Quentin Fallavier, France) was used as an internal control. Digitized images of the immunoblots obtained using a LAS-3000 imaging system (Fuji Photo Film Co., Ltd., Tokyo, Japan) were used for densitometry measurements with an image analyzer (UnScan It software, Silk Scientific Inc., Orem, Utah). 

Brain derived neurotrophic factor (BDNF) protein levels were measured in the brain using the BDNF EMax Immunoassay (ELISA E-Max, Promega, Madison, WI, USA). Protein preparations were incubated on a 96-well polystyrene ELISA plate previously coated with anti-BDNF monoclonal antibody. A standard curve was generated from serial dilutions of a human recombinant BDNF solution at 1 μg/mL. The captured neurotrophin was bound by a second specific anti-human BDNF polyclonal antibody, which was detected using a species-specific antibody conjugated to horseradish peroxidase (HRP). After removal of unbound conjugates, bound enzyme activity was assessed by chromogenic substrate for measurement at 450 nm by a microplate reader (Flex Station3, Molecular Devices). All assays were performed in duplicate.

### 4.6. Expression and Purification of Truncated DYRK1A (DYRK1A-ΔC)

The complementary DNA (cDNA) coding for the catalytic domain (residues 1–502) of rat DYRK1A (99.6% amino acid identity with human DYRK1A) was gifted by Prof. W. Becker (Aachen University, Aachen, Germany) and subcloned into pET28a plasmid to produce recombinant 6xHis-tagged DYRK1A catalytic domain (DYRK1A-ΔC). The pET28-Dyrk1a-ΔC plasmid was transformed into Escherichia coli cells (BL21) for production and purification of the enzyme. Briefly, transformed bacteria cells were grown at 37 °C for 4 h until OD was between 0.6–0.7 and further grown at 37 °C for 5 h in the presence of 0.5 mM isopropyl β-D-1-thiogalactopyranoside (IPTG). Cells were harvested by centrifugation at 6000× *g*, 4 °C for 10 min and the pellet was washed with PBS. The pellet was re-suspended in PBS supplemented with protease inhibitors, 1 mg/mL lysozyme and 0.1% Triton X-100. After 30 min incubation at 4 °C, the lysate was subjected to sonication on ice (6 min with pulse on 30 s/ off 30 s) and pelleted at 10,000× *g*, 4 °C for 30 min. The supernatant was incubated with His-select nickel resin (Sigma, Saint-Quentin Fallavier, France) for 2 h at 4 °C under agitation, and the resin was transferred into a column and washed with Tris-HCl 20 mM, pH 8, with 5 mM imidazole. Proteins were eluted in Tris-HCl 20 mM, pH 8, 300 mM imidazole, and dialyzed overnight against Tris-HCl 50 mM, pH 8, dithiothreitol (DTT) 10 mM, MgCl2 5 mM. Purified proteins were quantified with Bradford’s reagent (BioRad). Purity was assessed by sodium dodecyl sulfate-polyacrylamide gel electrophoresis (SDS-PAGE). Proteins were kept at −80 °C until use.

### 4.7. DYRK1A Inhibition Assays

*In vitro* measurement of DYRK1A kinase activity was carried out using an UFLC-based approach in combination with a fluorescent peptide substrate of DYRK1A. The fluorescent peptide substrate was derived from the sequence of the human transcription factor FKHR that is a physiological substrate of DYRK1A [46]. The peptide substrate, coupled to fluorescein by its N-terminal amino acid, has the following sequence: KISGRLSPIMTEQ. In vitro assays were performed in a 96-well plate, in a total volume of 50 µL consisting of DYRK1A kinase buffer (Tris-HCl 50 mM, pH 8, DTT 10 mM, MgCl2 5 mM), ATP 1 mM, peptide substrate 30 µM, and purified Dyrk1a-ΔC at 1 ng/µL. The enzymatic reaction was started by addition of the substrate and the mixture was incubated at 37 °C for 30 min. 50 µL of 15% HClO_4_ (*v*/*v*) was added to stop the reaction, and 25 µL of the mixture was analyzed by UFLC (Shimadzu) using a 5 µm Phenomenex Kinetex C18 150 × 4.6 mm column. Mobile phases used consisted of 0.12% trifluoroacetic acid (TFA) (solvent A) and acetonitrile in 0.12% TFA. Phosphorylated and unphosphorylated peptides were separated by isocratic flow (75% solvent A/25% solvent B) at a flow rate of 1.0 mL/min. The peptides were monitored by fluorimetry (excitation at 485 nm, emission at 530 nm) and quantified by integration of the peak absorbance area. A calibration curve established with different known concentrations of peptides was used for quantification [46]. For inhibition studies with green tea catechins, the compounds, dissolved in water, were added at four different concentrations (up to 100 µM) to the well prior to addition of ATP (1 mM final concentration). Determination of the mode of inhibition (competitive/non-competitive) of the compounds was obtained by carrying assays with different concentrations of ATP (200, 400 and 800 µM).

### 4.8. Quantification of FontUp^®^ Polyphenols in Plasma and Brain Samples

Brains were homogenized with 1.5 mm diameter beads for 2 min at 25 frequencies/second using a tissue lyser II (Qiagen, Hilden, Germany). The homogenates were centrifuged at 13,000 rpm for 10 min at 4 °C. The supernatants were transferred in a clear tube and mixed with the preservative solution as mentioned at one-tenth of its volume.

The catechin contents was analyzed as previously reported [47]. Two hundred microliters of plasma or brain samples homogenates were thawed at room temperature and 50 µL of 0.4 M sodium phosphate buffer (pH 7.4) were added. The sample was mixed with 10 µL of beta-glucuronidase (250 units) and 10 µL of sulfatase (10 units) and then incubated at 37 °C for 45 min. Then the plasma or brain samples were partitioned with ethyl acetate in the ratio of 1:1 (*v*/*v*). The liquid-liquid extraction was carried out three times for 1 min by shaking on vortex and the organic phases (upper phases) were pooled. The combined ethyl acetate phases were evaporated under vacuum to dryness. The dry residue was re-dissolved in 50% methanol in water and was kept at −80 °C before HPLC analysis. HPLC was performed with a LC Prominence (Shimadzu, Kyoto, Japan) coupled with a Coulochem III electrode array detector (ESA Inc., Bedford, MA, USA) using a reverse- Interchrom C18 column (3 µm, 150 × 2.1 mm) eluted with the eluent described below at a flow rate of 0.35 mL/min at 45 °C. HPLC analysis was performed using a linear gradient system with mobile phase A (20 mM NaH2PO4) and mobile phase B (acetonitrile). Linear gradient elution was performed as follows: 100% mobile phase A for 10 min; 55% mobile phase A for 30 min and return to 100% mobile phase A for 15 min. The eluent was monitored using the Coulochem-III electrochemical detector (5011 analytical cell) with potential settings at 600 mV (E1) and 700 mV (E2). Quantification was carried out using the external standard method. Catechin quantification was performed after data acquisition using an LC Solution (Shimadzu). 

### 4.9. Molecular Docking

The Dyrk1A 3D structure (PDB ID: 5AIK) was downloaded from the Protein Data Bank (PDB) [48]. The protonation states of all ionizable residues were computed using the PROPKA3 program [49]. The compounds, EGCG, ECG and EC, were built using Marvin software (Marvin 17.3.13.0, 2017, ChemAxon: http://www.chemaxon.com).

The Smina program [50], a fork of AutoDock Vina [51], with Vinardo scoring function [52] was used to perform the computational molecular docking. The compounds rotatable bonds were flexible during the docking procedure, and all the protein residues kept rigid. The Gasteiger atomic partial charges were assigned to the compounds and the protein. The compounds and the protein were converted in PDBQT format using the AutoDockTools package [53]. The entire protein surface was subjected to the molecular docking search. This procedure, called blind docking, enables the binding sites detection when no a priori knowledge is available on the localization of the binding site. The PyMOL software (PyMOL Molecular Graphics System, Palo Alto, CA, USA; http://www.pymol.org) was used for docking analysis and figures generation [54]. 

### 4.10. Data Analysis

Statistical analysis was done with the Student’s t-test using Statview software (Statview 3, Abacus Corporation, Baltimore, MD, USA). For multiple pairwise comparisons between genotypes and treatments*,* statistical analysis was done with two-way ANOVA followed by the Bonferroni/Dunnet post hoc test using Statview software. The results are expressed as mean ± SEM (standard error of the mean). Data were considered significant when *p* < 0.05. A p value of 0.06–0.10 was considered to indicate a statistical tendency given the small sample size. Correlations were determined by using Spearman’s rank correlation, as data were not normally distributed according to the Shapiro-Wilk test. 

## 5. Conclusions

Taken together, our pre-clinical findings demonstrate the safety of Fontup^®^ on hepatic and cardiac functions, and normalization of relevant plasma and neuronal biomarkers in mice overexpressing *Dyrk1A*. These results guide the dose to be used in future clinical studies both in adults and children with DS.

## Figures and Tables

**Figure 1 ijms-21-01404-f001:**
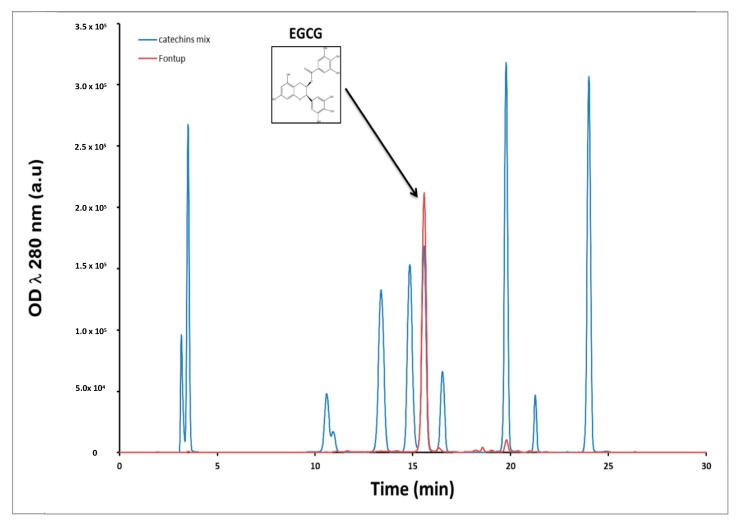
Chromatogram of the FontUp^®^ solution.

**Figure 2 ijms-21-01404-f002:**
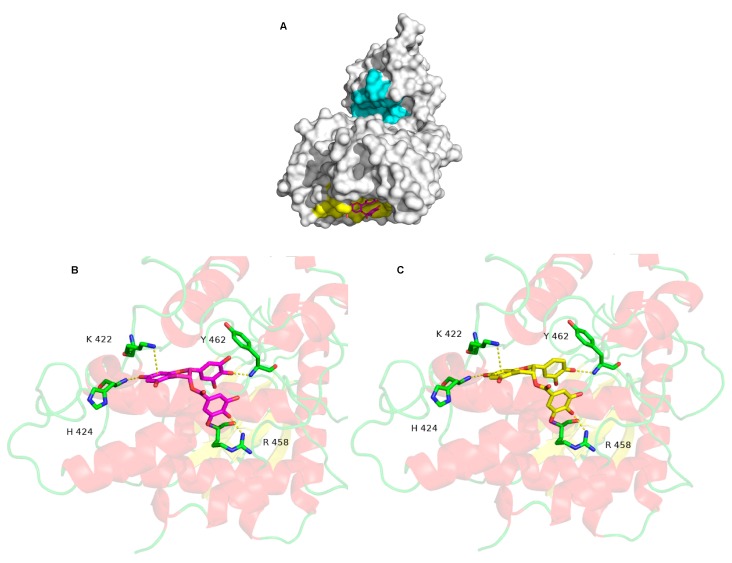
The molecular surface of DYRK1A is shown in white, in cyan, the catalytic site and in yellow, the most probable binding of EGCG located around the L 457 (**A**). EGCG is represented by stick models and its carbon atoms are colored in magenta. Detailed interactions between DYRK1A and EGCG (**B**), and ECG (**C**).

**Figure 3 ijms-21-01404-f003:**
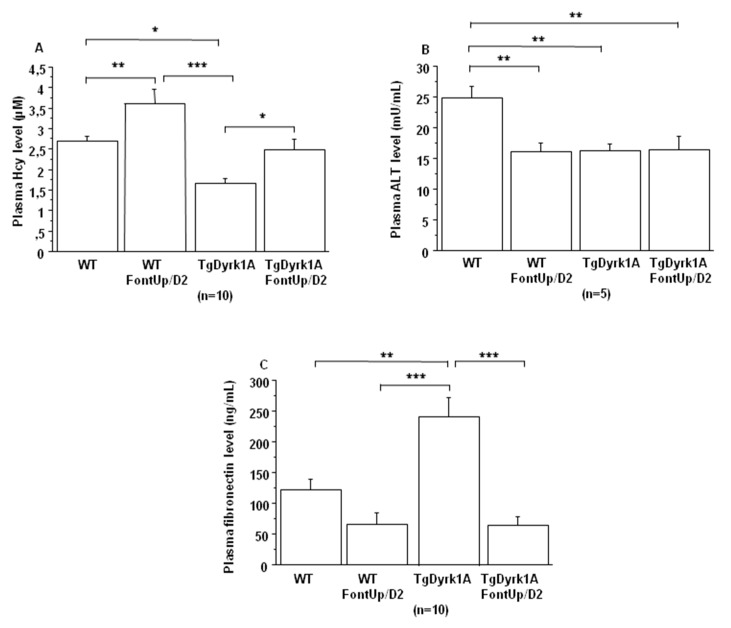
Comparison of plasma hcy (**A**), ALT (**B**) and fibronectin (**C**) levels obtained from WT and mice overexpressing *Dyrk1A* (TgDyrk1A) after oral gavage of D2 FontUp^®^ during two days (FontUp/D2). The values are mean ± SEM of n (number of mice) mice. Statistical analysis was done as mentioned in data analysis. * *p* < 0.05; ** *p* < 0.005; *** *p* < 0.0001.

**Figure 4 ijms-21-01404-f004:**
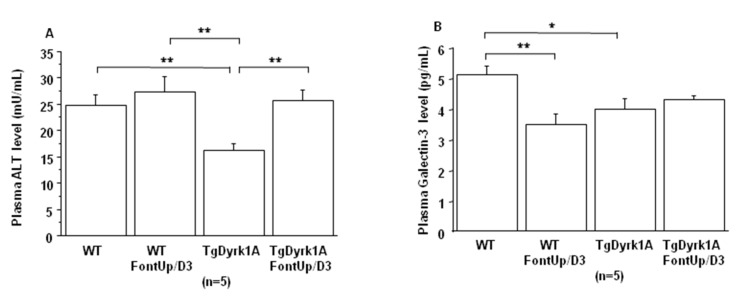
Comparison of plasma ALT (**A**) and Galectin-3 (**B**) levels obtained from WT and mice overexpressing *Dyrk1A* (TgDyrk1A) after oral gavage of D3 FontUp^®^ during eight days (FontUp/D3). The values are mean ± SEM of n (number of mice) mice. Statistical analysis was done as mentioned in data analysis. * *p* < 0.05, ** *p* < 0.005.

**Figure 5 ijms-21-01404-f005:**
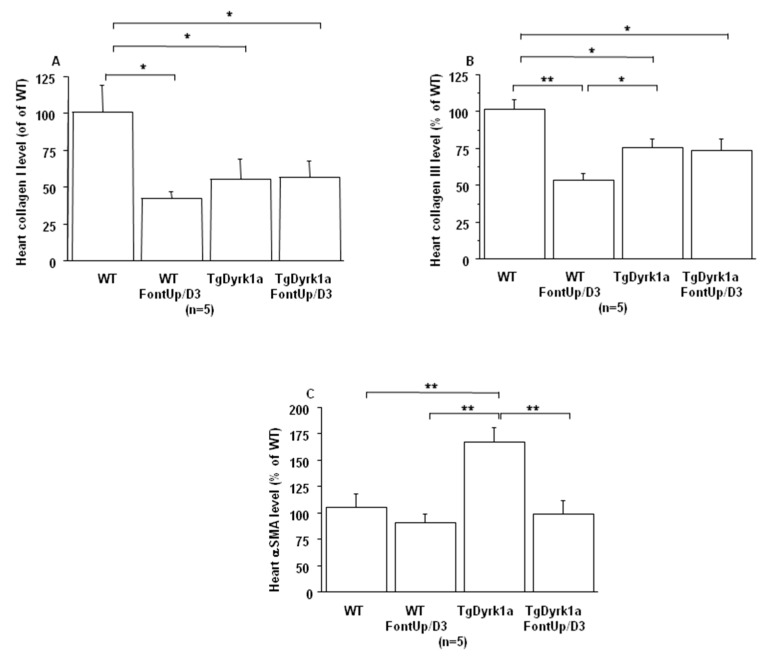
Comparison of relative heart collagen I (**A**), collagen III (**B**) and αSMA (**C**) protein levels based on slot blot analysis obtained from WT and mice overexpressing *Dyrk1A* (TgDyrk1A) after oral gavage of D3 FontUp^®^ during eight days (FontUp/D3). Values were obtained by normalization of images from collagen I, collagen III or αSMA to total proteins marked with Ponceau-S. The values are mean ± SEM of n (number of mice) mice normalized to the mean of WT mice without FontUp^®^. Statistical analysis was done as mentioned in data analysis. * *p* < 0.05; ** *p* < 0.005.

**Figure 6 ijms-21-01404-f006:**
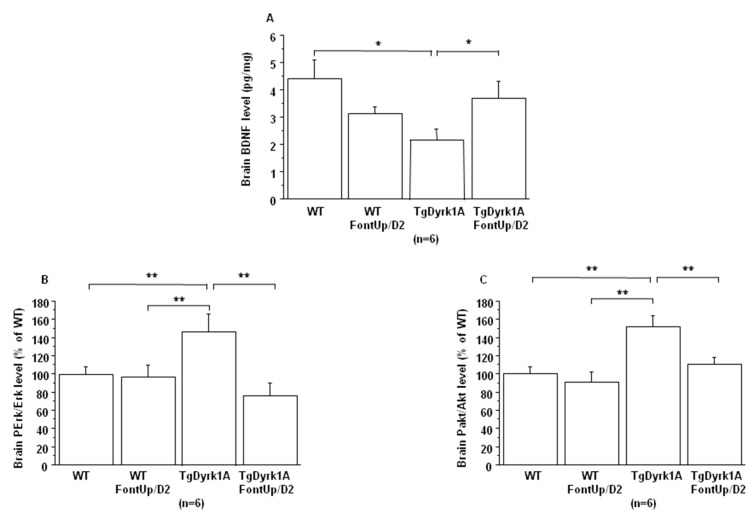
Comparison of brain BDNF level (**A**), and brain RTK activation (**B**, **C**) obtained from WT and mice overexpressing *Dyrk1A* (TgDyrk1A) after oral gavage of D2 FontUp^®^ during two days (FontUp/D2). BDNF levels were measured by ELISA, and Erk (**A**) and Akt (**B**) activation was analyzed by slot blot using antibodies specific to Perk or PAkt. After stripping, the membranes were rehybridized with anti Erk or anti Akt antibody for the control. Relative protein expression was determined by normalization of the density of images from PErk or PAkt with that of Erk or Akt of the same blot, and data were normalized to the mean of WT mice fed without FontUp^®^. The values are mean ± SEM of n (number of mice) mice. Statistical analysis was done as mentioned in data analysis. * *p* < 0.05; ** *p* < 0.005.

**Figure 7 ijms-21-01404-f007:**
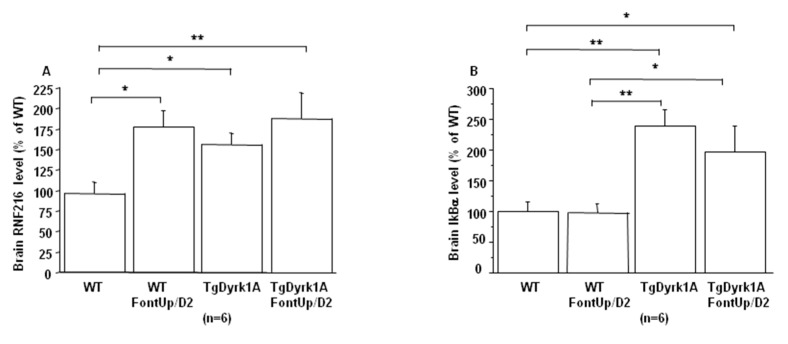
Comparison of brain RNF216 (**A**) and IkBα (**B**) level based on slot blot analysis obtained from WT and mice overexpressing *Dyrk1A* (TgDyrk1A) after oral gavage of D2 FontUp^®^ during two days (FontUp/D2). Values were obtained by normalization of images from RNF216 or IkBα to total proteins marked with Ponceau-S. The values are mean ± SEM of n (number of mice) mice normalized to the mean of WT mice fed without FontUp^®^. Statistical analysis was done as mentioned in data analysis. * *p* < 0.05; ** *p* < 0.005.

**Table 1 ijms-21-01404-t001:** DYRK1A-ΔC inhibition assay at different concentrations using HPLC-based assays.

Compound	Structure	DYRK1A-ΔC Remaining Activity at 0.1 µM (%)	DYRK1A-ΔC Remaining Activity at 1 µM (%)	DYRK1A-ΔC Remaining Activity at 10 µM (%)	DYRK1A-ΔC Remaining Activity at 100 µM (%)
ECG	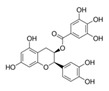	78.1 ± 3.6(*n* = 8)	27.1 ± 7.3(*n* = 8)	11.8 ± 4.2(*n* = 8)	7.6 ± 2.3(*n* = 8)
EGC	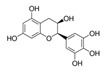	99.5 ± 3.2(*n* = 4)	93.6 ± 2.6(*n* = 4)	80.6 ± 11.4(*n* = 4)	81.4 ± 12.8(*n* = 4)
EGCG	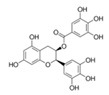	71.3 ± 4.6(*n* = 13)	16.3 ± 2.3(*n* = 13)	6.4 ± 1.1(*n* = 13)	3.5 ± 0.8(*n* = 13)
EC	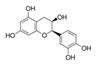	89.2 ± 9.5(*n* = 4)	84.1 ± 9.1(*n* = 4)	95.6 ± 8.1(*n* = 4)	88.7 ± 8.2(*n* = 4)

Inhibition of truncated DYRK1A activity (normalized in the absence of inhibitor, expressed as percentage) obtained with different polyphenol concentrations (0.1–100 μM) and 1 mM of ATP and 60 µM of peptide.

**Table 2 ijms-21-01404-t002:** DYRK1A inhibition assay at different concentrations using HPLC-based assays.

Compound	Structure	DYRK1A Remaining Activity at 0.1 µM (%)	DYRK1A Remaining Activity at 1 µM (%)	DYRK1A Remaining Activity at 10 µM (%)	DYRK1A Remaining Activity at 100 µM (%)
ECG	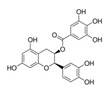	69.9 ± 3.9(*n* = 4)	12.5 ± 2.4(*n* = 4)	4.6 ± 0.9(*n* = 4)	2.3 ± 0.5(*n* = 4)
EGCG	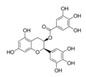	67.5 ± 9.4(*n* = 7)	10 ± 2(*n* = 7)	3.4 ± 0.6(*n* = 7)	2.1 ± 0.5(*n* = 7)

Inhibition of native DYRK1A activity (normalized in the absence of inhibitor, expressed as percentage) obtained with different polyphenol concentrations (0.1–100 μM) and 1 mM of ATP and 60 µM of peptide.

**Table 3 ijms-21-01404-t003:** Analysis of the inhibition of DYRK1A-ΔC activity by ECG and EGCG as a function of ATP concentrations.

Compound	DYRK1A-ΔC Remaining Activity With 200 µM ATP (%)	DYRK1A-ΔC Remaining Activity with 400 µM ATP (%)	DYRK1A-ΔC Remaining Activity with 800 µM ATP (%)
ECG (10 µM)	8.0 ± 0.8 (*n* = 4)	6.5 ± 0.9 (*n* = 4)	4.0 ± 0.5 (*n* = 4)
EGCG (10 µM)	11.0 ± 3.1 (*n* = 4)	9.6 ± 2.1 (*n* = 4)	5.4 ± 1.3 (*n* = 4)

Inhibition of DYRK1A-ΔC activity obtained with 10 μM of ECG and EGCG in the presence of different concentrations of ATP (200–800 μM) is shown. Assays were run for up to 60 min at 37 °C.

**Table 4 ijms-21-01404-t004:** Analysis of the inhibition of DYRK1A activity by ECG and EGCG as a function of ATP concentrations.

Compound	DYRK1A Remaining Activity with 200 µM ATP (%)	DYRK1A Remaining Activity with 400 µM ATP (%)	DYRK1A Remaining Activity with 800 µM ATP (%)
ECG (10 µM)	9.4 ± 1.4 (*n* = 4)	7.0 ± 1.1 (*n* = 4)	5.4 ± 0.9 (*n* = 4)
EGCG (10 µM)	6.3 ± 0.9 (*n* = 4)	5.1 ± 0.9 (*n* = 4)	4.0 ± 0.7 (*n* = 4)

Inhibition of DYRK1A activity obtained with 10 μM of ECG and EGCG in the presence of different concentrations of ATP (200–800 μM) is shown. Assays were run for up to 60 min at 37 °C.

**Table 5 ijms-21-01404-t005:** Plasma EGCG, hcy, GSH, ALT and Galectin-3 levels in WT mice that have received an oral gavage of low (D1), intermediate (D2) or high (D3) doses of FontUp^®^ during two days.

	WT CTL	WT D1	WT D2	WT D3
EGCG (nM)	0	28 ± 0.0003	63 ± 10.8	104 ± 11.2
hcy (µM)	2.7 ± 0.1	2.9 ± 0.3	3.6 ± 0.3 *	2.7 ± 0.2
GSH (µM)	66.4 ± 3.5	71.2 ± 2.6	75.2 ± 3.3 **	88.4 ± 2.7 ***
ALT (mU/mL)	27.4 ± 7.1	17.1 ± 5	14.9 ± 4.3 *	27.1 ± 4.9
Galectin-3 (pg/mL)	5.1 ± 0.3	5.2 ± 0.2	4.9 ± 0.25	5.1 ± 0.15

Values are mean ± SEM of 10 mice for each group. Statistical analysis was done as mentioned in data analysis. * *p* < 0.05; ** *p* < 0.02; *** *p* < 0.0003.

**Table 6 ijms-21-01404-t006:** Plasma and brain EGCG levels in WT mice and mice overexpressing *Dyrk1A* (TgDyrk1A) that have received on oral gavage of intermediate (D2) dose of FontUp during two days.

	WT	TgDyrk1A
Plasma EGCG (µM)	0.065 ± 0.023	0.036 ± 0.011
Brain EGCG (µM)	0.025 ± 0.007	0.050 ± 0. 01 ^$^
Kp	1.2 ± 0.33	3.6 ± 0.82

Values are mean ± SEM of five mice for each group. Statistical analysis was done as mentioned in data analysis. ^$^
*p* < 0.07.

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
