# Peer review of "Molecular Rescue of Dyrk1A Overexpression Alterations in Mice with Fontup® Dietary Supplement: Role of Green Tea Catechins"

_ijms, 2020, doi:10.3390/ijms21041404_

Round 1

Reviewer 1 Report

Manuscript ID: ijms-709891. Title: Molecular rescue of Dyrk1A overexpression alterations in mice with Fontup® dietary supplement: role of green tea catechins
In this manuscript to examine the effect of green tea catechins on Down syndrome-related  DYRK1A, the authors found that FontUp® can inhibit the enzyme activity of DYRK1A and improve some defects in the DYRK1A-overexpressing animals. The results are interesting, but revisions are required to improve the content on the basis of following comments.

Why were the changes in the mRNA expression levels of DYRK1Anot examined in WT and the DYRK1A-overexpressing animals by the Fontup® treatment?   Why were the changes in the protein levels of DYRK1Anot examined in the brain of WT and the DYRK1A-overexpressing animals by the Fontup® treatment?   Where is a source of DYRK1Aused in the inhibition assays? Leu 457 is not found in Fig. 2A (line 137, Fig. 2). ‘Abbreviations’(line 22) is incomplete. For example, truncated Dyrk1A, Perk, Pakt, and RNF are not listed. Although WT is defined, non-abbreviated description is often used, especially in Figure legends.  Explain gamma? catalytic site (line 139). gallic acid may be the gallic acid residue (line 296). Since ‘statistical analysis’is described in Materials and methods, a repetitive portion in Figure legends should be removed.

English should be improved. For example, sentences of lines 33, 38, 44, 55, 68, 98, 202, and 265.

Author Response

We thank the referee.

Why were the changes in the mRNA expression levels of DYRK1A not examined in WT and the DYRK1A overexpressing animals by the Fontup® treatment? Why were the changes in the protein levels of DYRK1A not examined in the brain of WT and the DYRK1A-overexpressing animals by the Fontup® treatment?

We did these two experiences and found no effect of FontUp® treatment (indicated now lines 311 and 312)

Where is a source of DYRK1A used in the inhibition assays?

DYRK1A-ΔC is home made (see materials and methods 4.6) and DYRK1A: DYRK1A native human protein, PV3785, Thermo Fisher Scientific, indicated line 112.

Leu 457 is not found in Fig. 2A (line 137, Fig. 2).

L457 allows to locate the pocket, but has no interaction (see line 180 and 207).

‘Abbreviations’(line 22) is incomplete. For example, truncated Dyrk1A, Perk, Pakt, and RNF are not listed.

Abbreviations have been completed.

Although WT is defined, non abbreviated description is often used, especially in Figure legends.

We corrected this point.

Explain gamma? catalytic site (line 139).

It’s a mistake.

gallic acid may be the gallic acid residue (line 296).

It’s gallic acid

Since ‘statistical analysis’is described in Materials and methods, a repetitive portion in Figure legends should be removed.

We corrected this point.

English should be improved. For example, sentences of lines 33, 38, 44, 55, 68, 98, 202, and 265.

English corrections have been made.

Reviewer 2 Report

This paper discusses the effect of regulation on several biomarkers related Dyrk1A on FontUp® administration and the activity of DYRK1A kinase on green tea catechins.

This manuscript is well written with interesting subject.

Need to make some English corrections.

Author Response

We thank the referee and english corrections have been made.